# Olive Pomace Extract Acts as a New Potent Ferroptosis Inhibitor in Human Cells

**DOI:** 10.3390/molecules30153095

**Published:** 2025-07-24

**Authors:** Edoardo Giuseppe Di Leo, Chiara Stranieri, Gianni Zoccatelli, Maria Bellumori, Beatrice Zonfrillo, Luciano Cominacini, Anna Maria Fratta Pasini

**Affiliations:** 1Department of Medicine, Section of Internal Medicine D, University of Verona, P.le L.A. Scuro, 37134 Verona, Italy; edoardogiuseppe.dileo@univr.it (E.G.D.L.); chiara.stranieri@univr.it (C.S.); luciano.cominacini@univr.it (L.C.); 2Department of Biotechnology, University of Verona, Strada Le Grazie 15, 37134 Verona, Italy; gianni.zoccatelli@univr.it; 3NEUROFARBA Department, University of Florence, via Ugo Schiff 6, Sesto Fiorentino, 50019 Firenze, Italy; maria.bellumori@unifi.it (M.B.); beatrice.zonfrillo@unifi.it (B.Z.)

**Keywords:** ferroptosis, glutathione, olive pomace extract, oxidative stress

## Abstract

The olive oil-production sector engages with the environment on multiple levels, and the valorization of olive pomace (OP) has emerged as a key strategy to improve the entire system’s sustainability. Numerous studies have investigated the biological effects of OP phenolic fraction for nutraceutical applications, highlighting its antioxidant properties. This study aimed to assess the effect of an OP extract (OPE) and its phenolic content on ferroptosis induced by RAS-selective lethal 3 (RSL3), an inhibitor of glutathione peroxidase 4. After characterization of OPE phenolic composition, its antioxidant properties were confirmed through the Fenton reaction assay. Subsequently, we examined the effect of OPE on ter-butyl hydroperoxide-induced ROS generation and lipid peroxidation in TPH-1 and HIECs cells and found that OPE reduced ROS and lipid peroxidation. RSL3 decreased the number of vital cells, which was associated with an elevation in ROS and lipid peroxidation, and a reduction in GSH. Interestingly, all these detrimental effects were reversed by OPE. Furthermore, OPE was also found to significantly increase GSH and the GSH/GSSG ratio per se. In conclusion, the fact that OPE decreases ROS and lipid peroxidation induced by RSL3 and augments GSH and cell viability suggests that OPE has potential as a ferroptosis inhibitor.

## 1. Introduction

Over the last three decades, the olive oil sector has undergone a process of continuous improvement, particularly in a more rigorous search for best practices in oil extraction and by-product processing in a circular economy perspective [1]. Although it contributes to human health and prosperity, the olive oil-production sector interacts with the environment on multiple levels, such as abiotic resource depletion, global warming, ecotoxicity, acidification, eutrophication, and others [1]. These impacts can vary significantly, depending on the olive-cultivation and oil-processing methods employed, as well as on the management of by-products and biomass residues [1]. In this context, the valorization of olive pomace (OP) has emerged as a key strategy to improve the overall sustainability of the system by enabling its use for bio-gas production to meet energy demands, collection of waste cooking oil for biodiesel manufacturing, animal feed production, and its valuable conversion into pharmaceutical ingredients or cosmetic products [1].

OP is the main solid residue remaining after olive oil production, consisting mainly of lignin, cellulose, hemicelluloses, residual olive oil, minerals, and phenolic constituents [2,3]. OP polyphenols have been the focus of recent research, given their positive impact on human health and the broad range of their biological effects [4]. They are primarily phenolic acids, phenolic alcohols, flavonoids, lignans, and secoiridoids [5]. Among them, hydroxytyrosol (OH-Tyr), tyrosol (Tyr), and their secoiridoidic precursors are considered the principal molecules responsible for the positive pharmacological effects associated with OP [5]. Given its phenolic composition, OP is remarkably similar to both olive oil and olive fruit and is regarded as an economic source of OH-Tyr derivatives. However, it is noteworthy that the biological effects of OP can be influenced by the generally low bioavailability of olive polyphenols due to the matrix-related extraction challenges [6], and therefore data on the bioavailability of olive polyphenols from oil cannot be extended to OP extracted with different matrices.

Ferroptosis is an iron-dependent type of non-apoptotic cell death that involves the accumulation of polyunsaturated fatty acid hydroperoxides (LOOH) resulting in oxidative damage of cell membranes [7]. Glutathione peroxidase 4 (GPX4) and system Xc- are regarded as the key signaling pathways connected with ferroptosis [7]. System Xc- is a heterodimeric amino acid transporter. The solute carrier family 7 member 11, an amino acid transporter, works to exchange L-cystine and L-glutamate [8]. Erastin, a small molecule working as a ferroptosis trigger, suppresses system Xc-, depleting glutathione (GSH), an event that indirectly inhibits glutathione peroxidase 4 (GPX4) [9]. The resulting cysteine lack hinders the synthesis of GSH and thereby leads to increased glutaminolysis [10]. Excessive glutaminolysis triggers mitochondrial tricarboxylic acid cycle activity and strongly amplifies mitochondrial respiration, resulting in hyperpolarization and increased production of reactive oxygen species (ROS) [10,11]. Superoxide is generated by the electron leakage from electron-transport chain complexes I and III and is subsequently converted into hydrogen peroxide (H_2_O_2_) through the activity of mitochondrial superoxide dismutase [11]. As reported by Dixon et al. [7], ROS can also be generated by NADPH oxidase 1 (NOX1) during erastin-induced ferroptosis. Unlike erastin, it is known that suppression of GPX4 function and the resulting accumulation of cellular LOOH that induces ferroptosis are caused by RAS-selective lethal 3 (RSL3) [7]. Recently, it has also been speculated that RSL3 can strongly inhibit the enzymatic activity of thioredoxin reductase 1 (TrxR1), which would shift the thioredoxin pool towards their oxidized forms [12]. The inhibition of the enzymatic activity of TrxR1 by RSL3, has in turn been reported to keep more protein disulfide isomerase (PDI), a member of the thioredoxin superfamily, in its oxidized form [12]. In this form, PDI facilitates nitric oxide synthase (NOS) dimerization/activation with accumulation of further ROS, nitric oxide, and generation of additional LOOH, which favor ferroptosis [13].

Dysregulated iron metabolism, characterized by the accumulation of labile iron ions in the cytoplasm, plays a central role in ferroptosis by catalyzing the Fenton reaction on H_2_O_2_ and LOOH from PUFAs, which leads to the production of highly reactive hydroxyl radicals (-OH) [7,9,12,13], which exponentially increase the generation of LOOH and cell death [7,9,14]. Normally, GPX4 transforms GSH into oxidized glutathione (GSSG) and converts the cytotoxic LOOH into the corresponding alcohols [6,8,14]. At the end, depletion of GSH and inhibition of GPX4 culminate in uncontrolled lipid peroxidation and cell death through ferroptosis [7,9,15,16,17]. The contribution of ferroptosis in pathological cell death connected with degenerative diseases, carcinogenesis, stroke, intracerebral hemorrhage, traumatic brain injury, ischemia–reperfusion injury, and kidney degeneration is increasingly being accepted [18].

Since OP is rich in bioactive compounds, especially phenolic compounds, its use in the treatments of situations characterized by abnormal oxidative stress like ferroptosis may be recommended. In this study, therefore, we evaluated the effect of OP extract (OPE) on RSL3-induced ferroptosis in immortalized THP-1 and in primary human intestinal epithelial (HIEC) cells.

## 2. Results

### 2.1. OPE OH-Tyr and Tyr Quantification

The HPLC chromatogram and the characterization of the phenolic compounds are presented in Appendix A and Table 1, respectively. The identification was based on a combination of HPLC-DAD-ESI-MS data, UV-Vis spectral analysis, and, where available, co-injection with authentic standards, according to our previous studies [19,20]. The phenolic profile of the OPE was similar to that presented in Fierri et al. (2024) [19], in which the same OP was used, with OH-Tyr and verbascoside identified as the main compounds. The sum of the total amounts of OH-Tyr and Tyr, quantified by HPLC after acid hydrolysis, was 4.96 ± 0.28 mg/g of dry OP. Given that the dry residue of the extract was 22.45%, the total OH-Tyr + Tyr content in the dry extract was 22.09 ± 1.25 mg/g. All subsequent experiments were conducted using the lyophilized OPE after solubilization, and the results are therefore expressed as µg/mL of dry OPE.

#### 2.1.1. Cell-Free Antioxidant Activity of OPE

Time- and dose-dependent antioxidant activity was evaluated using the Fenton reaction in a cell-free system (Figure 1A,B). In particular, the bar graph (Figure 1A) indicates the significant dose-response inhibitory effect of OPE on the Fenton reaction at 90 min, while Figure 1B is a line graph showing the fluorescence over time induced by the standard ammonium iron (II) sulfate (iron (II)) and by increasing concentrations of OPE added to iron (II).

#### 2.1.2. OPE Decreases ROS and Lipid Peroxidation Induced by TBHP in THP-1 and HIEC Cells

We first examined the dose-dependent effects of OPE on TBHP-induced ROS generation in THP-1 cells and HIECs. The preincubation of OPE with THP-1 and HIEC cells dose-dependently reduced the ROS production stimulated by TBHP (*p* from <0.05 to <0.01) (Figure 2A,B). Furthermore, compared to non-treated cells, TBHP-stimulated cells also showed increased levels of intracellular lipid peroxides (*p* from <0.001 to <0.0001), which were dose-dependently decreased by preincubation with OPE in both cell lines (*p* from <0.05 in THP-1 and *p* < 0.01 in HIEC cells) (Figure 2C,D).

#### 2.1.3. Effect of OPE in Reducing the RSL3-Induced Cytotoxicity in THP-1 and HIEC Cells

The incubation of OPE (from 50 to 150 µg/mL medium) with THP-1 did not cause any variation in the number of vital cells and proportion of apoptotic, pre-apoptotic, and necrotic cells (Figure 3A). In contrast, RSL3 (from 1 to 5 µM) determined a time- and dose-dependent significant decrease in the number of vital cells (*p* < 0.01), with a proportional increase in apoptotic, pre-apoptotic, and necrotic cells (from *p* < 0.001 to *p* < 0.01) (Figure 3B). Interestingly, this increase in apoptotic cells was significantly reversed (*p* < 0.0001) when cells were preincubated with OPE (Figure 2C). As expected, Lip-1, a potent ferroptosis inhibitor, abolished the RSL3-induced increase in apoptotic cells (Figure 3C). Furthermore, Figure 3D shows representative flow cytometric analyses of cell viability. 

#### 2.1.4. OPE Reduces ROS and Lipid Peroxidation Triggered by RSL3 in THP-1 and HIEC Cells

Since lipid peroxidation is a hallmark of ferroptosis, we then explored the effect of RSL3 on ROS formation in our experimental conditions. Our data show that when both cell lines were stimulated with RSL3, there was a significant elevation of intracellular ROS generation (Figure 4A), which was dose-dependently reduced by OPE (*p* from <0.001 to <0.0001) and Lip-1, a known ferroptosis inhibitor. RSL3 also dose-dependently significantly increased intracellular lipid peroxidation (*p* from *p* < 0.05 to *p* < 0.0001) both in THP-1 cells and in HIECs (Figure 4B,D). Interestingly, the preincubation of increasing concentrations of OPE was also able to counteract the cellular lipid peroxidation (*p* from <0.01 to <0.0001), as shown in Figure 4C,E, to a similar extent to Lip-1 (Figure 4C,D).

#### 2.1.5. OPE Increases GSH and GSH/GSSG Ratio in THP-1 and HIEC Cells Incubated with RSL3

Since OPE was demonstrated to revert intracellular oxidative stress and lipid peroxidation in our cellular models, we first evaluated whether overnight incubation of OPE could increase GSH, the key cellular antioxidant. Our results show that OPE (from 50 to 150 µg/mL) dose-dependently increased GSH levels (*p* < 0.0001) (Figure 5A). We then evaluated whether GSH concentrations were reduced in our cellular models of ferroptosis. Our results demonstrate that the RSL3-induced decrease in GSH concentration was significantly reversed when cells were preincubated with OPE or Lip-1 (Figure 5B). More intriguingly, the GSH/GSSG ratio, which was significantly affected by RLS3, was reverted entirely by OPE (Figure 5C). The fact that a similar effect was obtained with Lip-1, a known ferroptosis inhibitor, strengthens our results, indicating an anti-ferroptotic effect of OPE.

## 3. Discussion

One of the most important food processes in the Mediterranean basin is focused on olive cultivation and olive oil production [24]. Argun et al. [25] documented that the yearly olive oil production in the Mediterranean area is about 4 million tons, producing approximately 16 million tons of OP. Owing to their hydrophilic characteristics, phenolic compounds are abundant in OP, since only 2% are carried to olive oil during oil extraction. In comparison, 98% remain in the extract, including simple phenols like OH-Tyr and Tyr, and more complex molecules like polyphenols [26,27]. Among these compounds, secoiridoids and simple derivatives such as OH-Tyr and Tyr are the most valuable compounds in terms of antioxidant, antimicrobial, and health features [28]. Even if we did not evaluate the direct antioxidant effect of single components of OP in this study, the fact that OH-Tyr and Tyr had the highest concentrations in OPE may explain, at least in part, its strong antioxidant capacity, initially assessed using the Fenton reaction assay. This sensitive and versatile cell-free method simulates oxidative stress conditions in cells, evaluating the antioxidant capacity of complex mixtures in biological samples by measuring the production of hydroxyl radicals ·OH [29]. It is known that OH-Tyr works as a free radical scavenger and metal chelator [28]. The elevated antioxidant capacity of OH-Tyr is due to the presence of the o-dihydroxyphenyl moiety. It primarily functions as a chain breaker by donating a hydrogen atom to peroxyl radicals (ROO-). In this manner, the OH-Tyr-derived radical is less reactive due to the stabilization conferred by an intramolecular hydrogen bond within the phenoxy radical. Tyr has been shown to reduce lipid peroxidation products in cells, but to a significantly lesser extent than OH-Tyr [30]. This difference is likely due to Tyr’s higher O–H bond dissociation enthalpy (BDE) compared to OH-Tyr, making Tyr less prone to donate hydrogen atoms and thus less effective as a radical scavenger [31].

The results of this study also show that OPE was able to counteract the oxidative stress induced by TBHP in human cell models (Figure 2A,B). In our study, OPE decreased the ROS elevation caused by TBHP as monitored by using the fluorescent probe CellROX. Similarly, OPE reduced the lipid peroxidation induced by TBHP in THP-1 cells and HIECs in a time- and dose-dependent manner. Compared to what occurred in the cell-free system, OPE components’ strong free radical-scavenging activity may have had a leading role in inducing a powerful antioxidant effect [28]. Nevertheless, it has been suggested that olive oil phenolics may give supplementary antioxidant defense by enhancing the endogenous antioxidant systems and triggering various cellular signaling pathways [32,33,34,35].

Similarly to TBHP, RSL3 augmented ROS and lipid peroxidation as evaluated by BODIPY. The results of our study show that OPE time- and dose-dependently reduced ROS and lipid peroxidation in THP-1 and HIEC cells.

Furthermore, the results demonstrate that OPE time- and dose-dependently augmented the cellular concentration of GSH. The elevation of this molecule may represent an additional defense against the oxidative stress induced by TBHP and RSL3. GSH, as a carrier of an active thiol group in the form of a cysteine residue, acts as an antioxidant either directly by interacting with ROS and reactive oxygen/nitrogen species or by operating as a cofactor for various enzymes [36]. Even if, based on the present data, we cannot establish which component of the OPE extract is responsible for GSH elevation, it is interesting to underline that OH-Tyr has already been reported to increase the concentration of GSH in cultured cells subject to oxidative stress [37]. This effect was related to the rise of gamma-glutamyl-L-cysteine ligase and glutathione synthase activity [34], two key enzymes in the synthesis of GSH [38].

Taken together, these results demonstrate that OPE possesses a strong antioxidant capacity that may explain the powerful effect of OPE on cell viability during incubation with RLS3. It has been established that RSL3 functions as an inducer of ferroptosis [39]. In recent years, it has been widely accepted that RSL3 induces ferroptosis primarily by suppressing GPX4 function, leading to cellular ROS accumulation [13]. It has recently been reported that treating cells with RSL3 activates protein disulfide isomerase (PDI) by inhibiting thioredoxin reductase 1. The activated PDI then catalyzes nitric oxide synthase dimerization, followed by cellular NO, ROS, and lipid ROS accumulation, ultimately resulting in ferroptotic cell death [40]. This pathway further increases the cellular oxidative stress induced by RSL3 [40]. Several hypotheses have been suggested about the cell death mechanisms induced by ferroptosis. It is known that PUFAs are essential constituents of the cell membranes, and the chemical and geometric configuration of the lipid bilayer could be modified by heavy lipid peroxidation during ferroptosis [40]. In addition, peroxide cumulation could generate membrane pores and disrupt the barrier function, with the alteration of membrane permeabilization leading to cell death [40]. The fact that in our study, OP dose-dependently decreased ROS and lipid peroxidation in the cell membranes and in the end augmented cell viability may indicate that OP functions as a potent inhibitor of ferroptosis.

## 4. Materials and Methods

### 4.1. Reagents

PE Annexin V Apoptosis Detection Kit I, utilized for the determination of cell viability, was purchased from BD Biosciences (BD Pharmingen, Franklin Lakes, NJ, USA). Ter-butyl hydroperoxide (TBHP), Sodium L-ascorbate, and dihydrorhodamine 123 were purchased from Sigma-Aldrich (St. Louis, MO, USA). CellROX Deep Flow Cytometry Assay Kit and BODIPY 581/591 C11 (BODIPY) were purchased from Life Technologies (Grand Island, NY, USA). RAS-selective lethal 3 (RSL3) and Liproxstatin-1 (Lip-1) were purchased from MedChemExpress (Monmouth Junction, NJ, USA, Cat. No.: HY-100218A, Cat. No.: HY-12726). HEPES Buffered Saline was prepared with HEPES 20 mM and NaCl 150 mM at pH 7.4.

### 4.2. OP Phenolic Compound Extraction and Characterization

Pitted OP was collected in 2022 from Produttori Agricoli Gardesani olive mill (Caprino Veronese, Verona, Italy) and lyophilized through a Pascal Lio5P (Milan, Italy). OP was extracted as described by Fierri et al. [19]. Briefly, the dried OP was extracted twice with an EtOH:H_2_O mixture (8:2 *v*/*v*) at a solid-to-solvent ratio of 1:40, under magnetic stirring for 2 h, and then filtered. The obtained extract was defatted with n-hexane, then dried under vacuum at 40 °C for 1 h using a Rotavapor^®^ R-100 (Büchi Corporation, Flawil, Sankt Gallen, Switzerland). Part of the dried extract was re-dissolved in the same extraction solvent and analyzed by HPLC-DAD-MS using an HP 1260 MSD coupled with DAD and MSD detectors, and with an API/electrospray interface (Agilent Technologies). The analytical conditions are the same as reported in our previous study [20]. A Poroshell 120 EC-C18 column (150 × 3.0 mm, 2.7 µm ps; Agilent, Palo Alto, CA, USA) working at 26 °C was employed with a mobile phase (flow 0.4 mL/min) constituted by CH_3_CN (A) and H_2_O (B, pH 3.2, formic acid). The following multistep linear gradient was applied: solvent B varied 95–60% in 40 min, stayed at 60% for 5 min, varied 60% to 0% in 5 min, stayed at 0% for 3 min, and finally returned to 95% in 2 min. The following ESI parameters were used: capillary voltage 3500 V; drying gas flow and temperature, 12.0 L/min, 350 °C; nebulizer pressure 1811 Torr. Full spectrum scan (range 100–1200 Th) was applied for acquisition in negative ion mode (fragmentor voltage = 150 V). The following calibration curves at 280 nm are used for the quantitation of phenolic compounds: tyrosol (linearity range 0–1.198452 µg; R^2^ = 0.9999) was used for tyrosol and their derivatives; hydroxytyrosol (0–6.850 µg; R^2^ = 0.9999); luteolin (0–2.1093 µg, R^2^ = 0.9999) for luteolin and flavonoids 1 and 2; verbascoside (0–6.368 µg, R^2^ = 0.9998) was used for β-OH-acteoside isomers 1 and 2, the putative verbascoside isomer, cafselogoside, comselogoside, and verbascoside; caffeic acid (0–0.96 µg, R^2^ = 0.9989) [20]. Acid hydrolysis of the extract was also performed before HPLC analysis to quantify the total Tyr and OH-Tyr content. To this purpose, 300 μL of extract were combined to 300 μL of 1 M H_2_SO_4_ solution and treated for 2 h at 80 °C. After cooling at room temperature, the mixture was diluted with 400 μL of H_2_O and centrifuged for 5 min at 13,000× *g*. The obtained supernatant was analyzed by HPLC. The rest of the extract was lyophilized and used for subsequent experiments.

### 4.3. Cell-Free Antioxidant Capacity of OPE

OPE cell-free antioxidant capacity was assessed through a Fenton reaction assay [29]. The Fenton reaction was performed as previously described with a few modifications [41]. Triplicates of 5 µL of OPE at increasing concentrations (from 50 µg/mL to 150 µg/mL) were transferred to 96-well plates and mixed with 195 µL of HBS buffer containing dihydrorhodamine 123 50 µM and sodium ascorbate 20 µM. As a positive control for hydroxyl radical production, the standard ferrous ammonium sulfate Fe(NH_4_)SO_4_ was used at 3.55 µM. Immediately following reagent addition, fluorescence was measured at 42 °C every 3 min over a total period of 90 min using a fluorescence plate reader (Fluoroskan Ascent, Thermo Electron Corporation, Vantaa, Finland; excitation 485 nm; emission 520 nm).

### 4.4. Cell Cultures

Human monocytic leukemia cells THP-1 (AddexBio, San Diego, CA, USA) were grown as previously described [42]. Briefly, the cells were expanded in RPMI 1640 medium (AddexBio, San Diego, USA) supplemented with 10% of FBS, 1% of penicillin-streptomycin solution, 0.1% of amphotericin B, 0.1% of tylosin, and 0.5% of heparin at 37 °C in a humified atmosphere containing 5% CO_2_. The selection of the above cell line was made based on extant evidence of their high reliability in evaluating oxidative stress and antioxidant signaling pathways [43,44]. HIECs (CellBiologics, Chicago, IL, USA) were grown in a human epithelial cell medium (CellBiologics, Chicago, IL, USA) supplemented with 5% of FBS, 1% of antibiotic-antimycotic solution, 0.1% of hydrocortisone, and 0.1% epithelial growth factor at 37 °C in a humified atmosphere containing 5% CO_2_.

Endotoxin contamination of cell culture was routinely excluded with the chromogenic Limulus amebocyte lysate assay.

### 4.5. Cell Viability Assay

It is widely acknowledged that the quantification of cell viability serves as the basis for a plethora of in vitro assays that assess cellular responses to external stimuli [45,46]. Therefore, the initial step in the experimental process involved the assessment of cell viability. This assessment was performed using flow cytometry, a highly informative test that distinguishes between viable, necrotic, and apoptotic cells. Cell lines were cultured at 1 × 10^6^ cells/mL in 24-well plates for this assay. Then, the medium was replaced and increasing concentrations of OPE (from 50 to 150 µg/mL) were incubated overnight. For dose- and time-dependent experiments, RSL3 was incubated for 6 and 24 h with increasing concentrations (from 1 to 5 mM of RSL3) in both cell lines. Cellular viability was evaluated with a PE Annexin V Apoptosis Detection Kit (BD Biosciences, NJ, USA) as previously reported [47,48,49]. The fluorescence intensity of cells per sample was measured by flow cytometry using the FACS BD Canto cytofluorometer. A minimum of 10,000 cells were analyzed by flow cytometry, and quantitative analysis was performed in FlowJo (BD Biosciences, Franklin Lakes, NJ, USA). All the assays were performed in triplicate.

### 4.6. Intracellular ROS Measurement

CellROX Deep Red Flow Cytometry Assay Kit (Molecular et al., Carlsbad, CA, USA) was used for the determination of intracellular ROS formation [44]. The CellROX Deep Red reagent, which is cell-permeable, is essentially non-fluorescent when it is in its reduced state. However, upon oxidation, it exhibits a strong fluorogenic signal, making it a reliable indicator of ROS in live cells [50]. Cells were seeded in 24-well plates at a density of 5 × 10^5^ cells/mL. To explore the effect of OPE on counteracting oxidative stress, increasing concentrations (from 50 to 150 μg/mL) of OPE were added to the cells overnight. Post incubation, cells were rinsed with phosphate-buffered saline (PBS) and exposed to 200 μM tert-butyl hydroperoxide (TBHP) for 45 min at 37 °C to serve as a positive control for oxidative stress. After incubation of the cells with TBHP, the CellROX Deep Red reagent was added to the cells for 45 min at 37 °C at a final concentration of 500 nM, and then immediately analyzed by flow cytometry.

### 4.7. Determination of Cellular Lipid Peroxidation

Lipid peroxidation was measured using C11-BODIPY (581/591), as detailed in reference [51]. This reagent serves as a sensitive fluorescent reporter for lipid peroxidation; upon oxidation in live cells, the fluorescence of the fluorophore shifts from red to green, providing a quantifiable metric for lipid peroxidation when analyzed by flow cytometry [52]. To assess the impact of OPE on lipid peroxidation, increasing concentrations of OPE (ranging from 50 to 150 µg/mL) were added to THP-1 cells and HIECs overnight, before the addition of TBHP or RSL3. Following this, the cells were stained with 2 µM BODIPY for 30 min at 37 °C and promptly analyzed by flow cytometry. Fluorescence was measured at two distinct wavelengths: one at an excitation/emission of 581/591 nm (PE filter set) for the reduced dye and the other at 488/510 nm (FITC filter set) for the oxidized dye. The results were presented as the signal ratio from the 590 nm to the 510 nm channel.

### 4.8. Ferroptosis Induction and the Effect of OPE

We first set up some preliminary time- and dose-dependent experiments to find the best conditions for inducing ferroptosis in both THP-1 cells and HIECs using erastin and RLS3. Because our preliminary results indicate that THP-1 cells were insensitive to erastin, as previously reported [53] we chose to generate ferroptosis by using different concentrations of RSL3 (from 1 to 5 µM in THP-1 and from 1 to 10 µM in HIECs) for 6 or 24 h.

To evaluate the impact of OPE on RSL3-induced ferroptosis, increasing concentrations of OPE (from 50 to 150 µg/mL) were added overnight to THP-1 cells and HIECs before the addition of RSL3. As control, we employed Lip-1 (2 µM), a known potent inhibitor of ferroptosis [54].

### 4.9. Measurement of GSH and GSSG

We finally explored the effect of OPE on cellular GSH and GSSG in THP-1 cells and HIECs. Cells were preincubated overnight with either Lip-1 (2 µM) or OPE (50, 100, and 150 µg/mL) followed by stimulation with RSL3. GSH and GSSG were measured following the method of Enomoto et al. [55] by using an Agilent Technologies 1260 Infinity System (Agilent Technologies, Santa Clara, CA, USA) consisting of an autosampler (G1367E), pump (G1311B), and column compartment (G1316A) connected to an Agilent Technologies 6460 triple quadrupole mass spectrometer (Agilent Technologies, Santa Clara, CA, USA).

### 4.10. Statistical Analysis

Data were presented as mean ± standard deviation (SD) for a minimum of three independent experiments. Data were compared using one-way analysis of variance (ANOVA) followed by Dunn’s post hoc test and significance differences at *p* < 0.05 were obtained using the software GraphPad Prism (version 10.4.2, GraphPad software, Boston, MA, USA).

## 5. Conclusions

In conclusion, the results of this study show that OPE dose-dependently decreased RSL3-induced ferroptosis by inhibiting ROS generation and lipid peroxidation. Furthermore, OPE was also found to increase the concentrations of GSH. This compound acts as an antioxidant directly by interacting with ROS and reactive oxygen/nitrogen species or operating as a cofactor for various enzymes [37]. Since the contribution of ferroptosis in pathological cell death connected with degenerative diseases, carcinogenesis, stroke, intracerebral hemorrhage, traumatic brain injury, ischemia–reperfusion injury, and kidney degeneration is increasingly being accepted [20], accordingly, OP should also be recognized as a potential adjunctive therapy in these pathological situations.

## Figures and Tables

**Figure 1 molecules-30-03095-f001:**
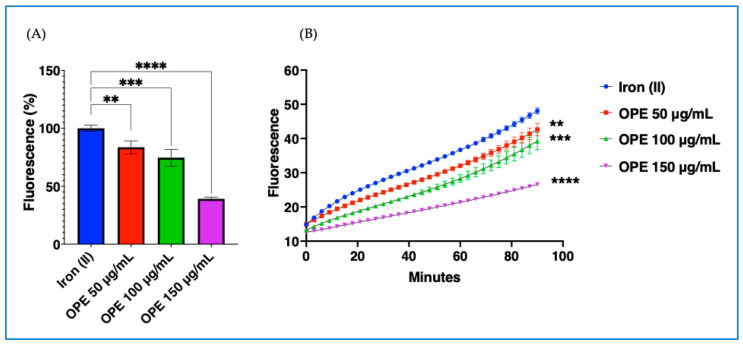
**Radical-scavenging effect of the olive pomace extract (OPE) on the ascorbate-driven Fenton reaction.** (**A**) Dose-responsive inhibitory effect of the OPE on the Fenton reaction at 90 min. (**B**) Fluorescence over time induced by the standard iron (II) and by OPE. Data represent the mean ± SD of measurements performed in triplicate in three different experiments. ** *p* < 0.01; *** *p* < 0.001; **** *p* < 0.0001 decrease vs. iron (II).

**Figure 2 molecules-30-03095-f002:**
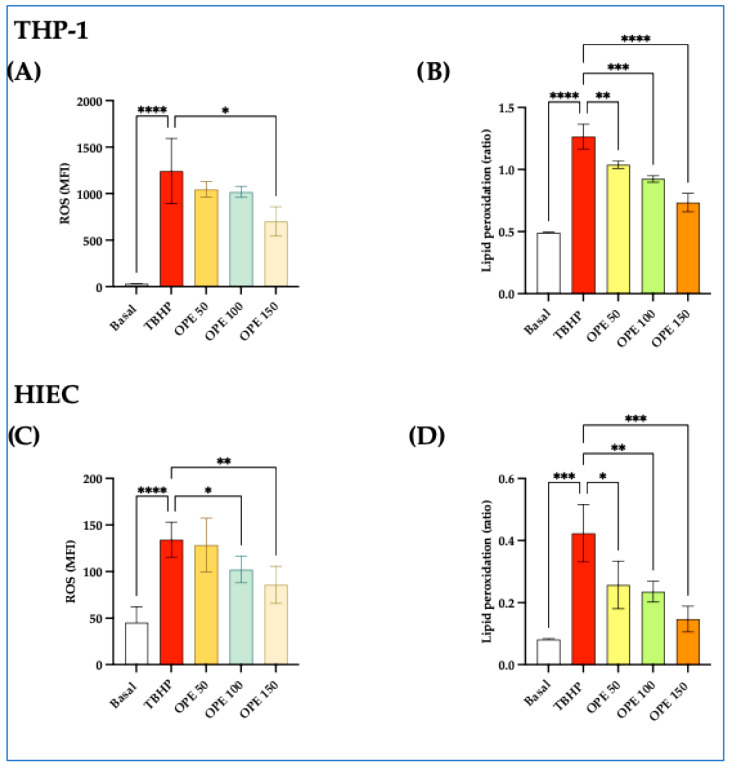
**Olive pomace extract (OPE) decreases intracellular reactive oxygen species (ROS) levels and lipid peroxidation in THP-1 and HIEC cells.** (**A**) OPE-induced decrease in ROS levels in THP-1 cells, expressed as mean fluorescence intensity (MFI). (**B**) Dose-dependent effect of OPE in mitigating lipid peroxidation, expressed as the ratio of oxidized dye fluorescence (510 nm) to reduced dye fluorescence (590 nm) in THP-1 cells. (**C**,**D**) OPE-induced reduction in ROS and lipid peroxidation in HIECs. Data are presented as mean ± SD. Cells were preincubated overnight with increasing concentrations of OPE (50–150 µg/mL) prior to stimulation with 200 µM of TBHP for 45 min. Statistical significance was determined by one-way ANOVA followed by Dunn’s post hoc test; * *p* < 0.05, ** *p* < 0.01, *** *p* < 0.001, **** *p* < 0.0001.

**Figure 3 molecules-30-03095-f003:**
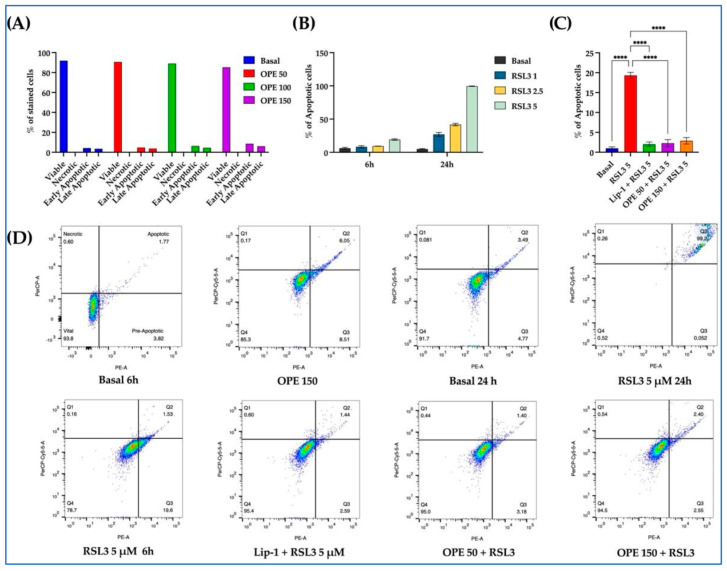
**Olive pomace extract (OPE) preserves cell viability and reduces RSL3-induced apoptosis in THP-1 cells.** Cells were preincubated overnight with increasing concentrations of OPE (50–150 µg/mL) or Lip-1 (2 µM) before stimulation with RSL3. (**A**) Dose-dependent effect of OPE on cell viability after overnight incubation. (**B**) Time- and dose-dependent induction of apoptosis after incubation with RSL3 (1–5 µM) for 6 or 24 h. (**C**) Protective effect of OPE preincubation against RSL3-induced apoptosis at 24 h. (**D**) Representative flow cytometric analyses of PE Annexin V staining in THP-1 cells. Data are presented as mean ± SD. Statistical significance was assessed by one-way ANOVA followed by Dunn’s post hoc test; **** *p* < 0.0001.

**Figure 4 molecules-30-03095-f004:**
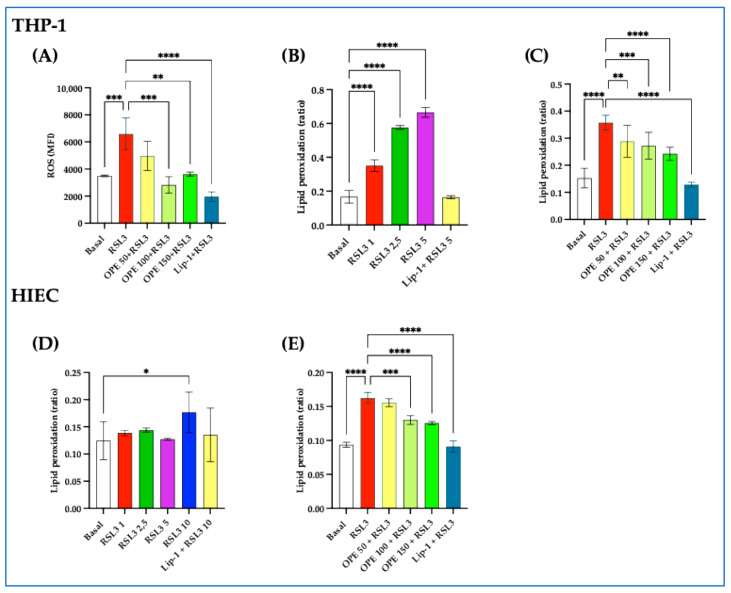
**Olive pomace extract (OPE) protects against RSL3-induced oxidative stress and lipid peroxidation in THP-1 and HIEC cells.** (**A**) Intracellular ROS reduction induced by OPE and Liproxstatin-1 (Lip-1) in THP-1 cells following RSL3 stimulation, expressed as mean fluorescence intensity (MFI). (**B**) Dose-response effect of RSL3 on lipid peroxidation in THP-1 cells. (**C**) The dose-dependent effect of OPE and Lip-1 in mitigating lipid peroxidation in THP-1 cells is expressed as the ratio of oxidized dye fluorescence (510 nm) to reduced dye fluorescence (590 nm). (**D**) Dose-response effect of RSL3 on lipid peroxidation in HIECs. (**E**) Dose-dependent effect of OPE and Lip-1 in reducing lipid peroxidation in HIECs. Cells were preincubated overnight with increasing concentrations of OPE (50–150 µg/mL) and Lip-1 (2 µM) before stimulation with RSL3 (5 µM) for 6 h. Data are presented as mean ± SD. Statistical significance was assessed by one-way ANOVA followed by Dunn’s post hoc test; * *p* < 0.05; ** *p* < 0.01; *** *p* < 0.001; **** *p* < 0.0001.

**Figure 5 molecules-30-03095-f005:**
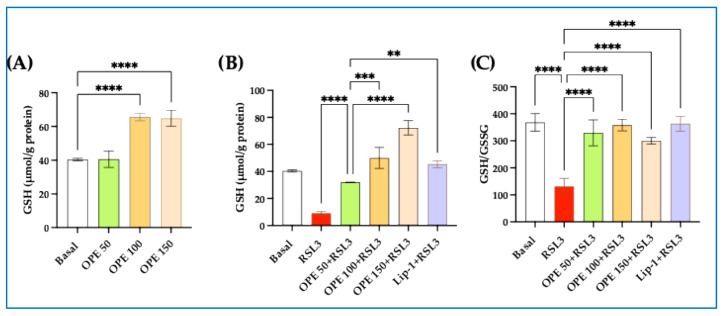
**Effect of olive pomace extract (OPE) and RSL3 treatments on intracellular-reduced glutathione (GSH) and GSH/oxidized glutathione (GSSG) ratio levels.** (**A**) Dose-response effect of OPE on GSH. (**B**) Inhibitory effect of RSL3 on GSH and dose-response effect of preincubation of OPE and Liproxstatin-1 (Lip-1) on cellular GSH. (**C**), GSH/GSSG ratio after overnight treatment with either OPE or Lip-1. Cells were incubated overnight with Lip-1 (2 µM) or increasing concentrations of OPE (from 50 to 150 µg/mL), followed by stimulation with RSL3 5 µM. Data are presented as mean ± SD. Statistical significance was assessed through one-way ANOVA followed by Dunn’s post hoc test; ** *p* < 0.01; *** *p* < 0.001; **** *p* < 0.0001.

**Table 1 molecules-30-03095-t001:** Retention time, maximum wavelength absorbance, MS fragmentation profiles, and amounts (mg/g) of phenolic compounds identified in lyophilized OP.

Peak Number	Rt (min)	λ max	MS Peaks	Molecule	Ref.	Mean	SD
**1**	6.98	276	315–153	OH-tyr hexose	[21,22,23]	0.71	0.12
**2**	7.08	280	153–123	OH-tyr	STD	1.14	0.11
**3**	10.21	276		tyr	STD	0.24	0.02
**4**	11.36	270	377–151	Putative tyr derivate		0.13	0.01
**5**	12.34	262–292	565–467	Putative tyr derivate		0.15	0.02
**6**	13.07	300–322	179	Caffeic acid	STD	0.12	0.02
**7**	17.09	288–330	639	OH-acteoside 1	[22]	0.13	0.02
**8**	17.32	288–330	639	OH-acteoside 2	[22]	0.11	0.02
**9**	18.95	274–350	947–473	Flavonoid 1		0.08	0.01
**10**	20.04	254–354	609	Flavonoid 2	[22,23]	0.10	0.01
**11**	21.37	288–330	623	verbascoside	STD	1.42	0.15
**12**	22.78	288–330	623	verbascoside iso	[22]	0.09	0.00
**13**	24.25	290–328	551	cafselogoside	[22,23]	0.24	0.04
**14**	26.83	226–310	535	comselogoside	[21,22,23]	0.28	0.03
**15**	30.45	252–350	285	luteolin	STD	0.306	0.030
				* total tyr + OH-tyr		4.96	0.28

*** Quantified upon hydrolysis.**

## Data Availability

Data are available on request from the corresponding author.

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
