# Peer review of "Olive Pomace Extract Acts as a New Potent Ferroptosis Inhibitor in Human Cells"

_molecules, 2025, doi:10.3390/molecules30153095_

Round 1
Reviewer 1 Report
Comments and Suggestions for Authors
The manuscript is interesting, and the results presented regarding the benefits of OPE for cellular life are proven.
However, there are some aspects that could certainly be improved, because from a scientific standpoint, they are improvable. For example, the chromatogram presented in the ESI lacks scientific rigour. It is a chromatogram of an OPE extract in which some chromatographic peaks appear that are assigned ‘supposed’ chemical structures of dubious certainty.
In order to identify them, it would be necessary to have the corresponding pure patterns, commercial or otherwise, and confirm the presence of these compounds by characterising them according to their retention indices k'.
Another alternative would be to perform the separation by HPLC-MS and, from the mass spectra, carry out an ‘identification’ or assignment of the most probable structures.
1.- How have the compounds that are separated and shown in the chromatogram in Figure S1 been identified and patterns injected?
2.- A description of the chromatographic procedure is needed, as in the reference 39 (Food Chem) offprints apply it to other samples and the extraction procedure is different. In this reference (39) the authors inject standards and obtain calibration curves to quantify, here they should do the same. Furthermore the characteristics of the instrumentation HPLC-DAD, column (chemistry of the stationary phase, particle diameter, length column, inner diameter of the column...), mobile phase composition, flow rate..., should be included.
3.- In the chromatogram, are compounds 4 and 5 appearing as two TYR derivates, ere they different? What about the case of compounds 7 and 8 called OH-acteoside 1 and OH-acteoside 2? Do you know their chemical structures?
If they do not inject standard compounds such as in the 27 and 39 references in the manuscript, they should justify in some way the reasons that lead them to think that the compounds that appear in the chromatogram are the ones they claim. For example, the authors could look for the log P of the compounds that they allude to and they can study if any correlation can be established with the order of elution and the chemical structure an physicochemical properties of the compounds proposed to be present in the extracts.
4.-Cell-free antioxidant capacity of OPE : The authors claim to have determined the antioxidant capacity by means of the Fenton reaction assay. They say that antioxidant capacity is determined by measuring fluorecence response. Can they specify what methodology they are using? will it be the ABTS assay? Please say so that if they want to reproduce the experimental conditions they can do so, not just say "few" modifications with respect to 21 and 40. Can the authors explain the meaning of "few"?
5.- Intracellular ROS Measurement: Even though it is a standardised procedure (kit), a detailed description of the experimental procedure would be desirable, especially in connection with the results.
6.- Determination of cellular lipid peroxidation: If you are using the fluorescence of the bodipy probe for the determination of cellular lipid peroxidation, please indicate the wavelengths of the additional fluorescence filters but do not allude to fluorescein isothiocyanate because this probe are not employed in this assay. Precise the range of wavelengths for excitation and emission conditions.
7.- Conclusions:The conclusions describe the beneficial effects of OPE on cell life by reducing ROS generation and lipid peroxidation. Given that no specific compounds that are present in the extracts of OPE, I would recommend avoiding the inclusion of experiments that are not properly designed and do not contribute anything (HPLC). Thus, the discussion refers in general terms to the role of polyphenols and specifically to TYR, but without being able to state anything concrete, except to refer to its role as a scavenger of metals and oxidising species. Please avoid speculations.
Author Response
The manuscript is interesting, and the results presented regarding the benefits of OPE for cellular life are proven.
However, there are some aspects that could certainly be improved, because from a scientific standpoint, they are improvable. For example, the chromatogram presented in the ESI lacks scientific rigour. It is a chromatogram of an OPE extract in which some chromatographic peaks appear that are assigned ‘supposed’ chemical structures of dubious certainty.
In order to identify them, it would be necessary to have the corresponding pure patterns, commercial or otherwise, and confirm the presence of these compounds by characterising them according to their retention indices k'.
Another alternative would be to perform the separation by HPLC-MS and, from the mass spectra, carry out an ‘identification’ or assignment of the most probable structures.
We thank the reviewer for raising this important concern. We would like to clarify that the identification of the peaks was not speculative: it was based on a combination of HPLC-DAD-ESI-MS data, UV-visible spectral analysis, and, where available, co-injection with authentic standards. This methodology is in line with what we have previously reported in our earlier studies and follows commonly accepted practices in the analysis of complex natural extracts. To address the reviewer’s concern and improve transparency, we have now explicitly detailed the identification criteria used for each compound in the revised version of the manuscript, including MS fragmentation patterns, UV maxima, and retention time comparisons with standards or literature data. We trust that this additional information will clarify the scientific basis for the peak assignments and strengthen the rigor of the chromatographic analysis presented.
1.- How have the compounds that are separated and shown in the chromatogram in Figure S1 been identified and patterns injected?
We replied to this question in the previous answer. A full description of the methodological part has been included (under materials and methods-OP phenolic compounds extraction and characterization, page 10 from line 5 to line 26).
2.- A description of the chromatographic procedure is needed, as in the reference 39 (Food Chem) offprints apply it to other samples and the extraction procedure is different. In this reference (39) the authors inject standards and obtain calibration curves to quantify, here they should do the same. Furthermore the characteristics of the instrumentation HPLC-DAD, column (chemistry of the stationary phase, particle diameter, length column, inner diameter of the column...), mobile phase composition, flow rate..., should be included.
We thank the reviewer for this important remark. In the revised version of the manuscript, we have now included a detailed description of the chromatographic method, including all relevant parameters: column specifications, mobile phase composition, gradient program, flow rate, and detector settings. Although our initial intention was to keep the focus on the biological activity of the extract, assuming that the citation of new reference 20, would suffice for methodological background, as mentioned above, we agree that providing the complete chromatographic information enhances the reproducibility and scientific value of the work.
3.- In the chromatogram, are compounds 4 and 5 appearing as two TYR derivates, ere they different? What about the case of compounds 7 and 8 called OH-acteoside 1 and OH-acteoside 2? Do you know their chemical structures?
If they do not inject standard compounds such as in the 27 and 39 references in the manuscript, they should justify in some way the reasons that lead them to think that the compounds that appear in the chromatogram are the ones they claim. For example, the authors could look for the log P of the compounds that they allude to and they can study if any correlation can be established with the order of elution and the chemical structure an physicochemical properties of the compounds proposed to be present in the extracts.
We thank the reviewer for this detailed comment. Regarding compounds 4 and 5, they were assigned as two putative distinct tyrosol derivatives based on differences in retention time, UV absorbance profiles, and MS/MS fragmentation patterns. Compounds 7 and 8 were assigned as OH-acteoside isomers, which are well-documented components of olive pomace extracts, as also reported in the recent literature (e.g., https://doi.org/10.1016/j.jfca.2024.106203). Although authentic standards were not available for these specific compounds, the identification was supported by MS fragmentation patterns, which in the case of OH-actosides matched also literature data, and UV-Vis spectra.
4.-Cell-free antioxidant capacity of OPE: The authors claim to have determined the antioxidant capacity by means of the Fenton reaction assay. They say that antioxidant capacity is determined by measuring fluorecence response. Can they specify what methodology they are using? will it be the ABTS assay? Please say so that if they want to reproduce the experimental conditions they can do so, not just say "few" modifications with respect to 21 and 40. Can the authors explain the meaning of "few"?
We have modified the text with additional information about the method (
5.- Intracellular ROS Measurement: Even though it is a standardised procedure (kit), a detailed description of the experimental procedure would be desirable, especially in connection with the results.
We thank the reviewer for the suggestion and have modified the text with a more detailed description of the method (under materials and methods- page 11 from line 2 to line 4)
6.- Determination of cellular lipid peroxidation: If you are using the fluorescence of the bodipy probe for the determination of cellular lipid peroxidation, please indicate the wavelengths of the additional fluorescence filters but do not allude to fluorescein isothiocyanate because this probe are not employed in this assay. Precise the range of wavelengths for excitation and emission conditions.
The bodipy fluorescent probe works differently than “common” fluorescent probes. This probe is an undecanoic acid that localizes in the cell membrane and has a polyunsaturated butadienyl portion that emits fluorescence. When in basal conditions, i.e., when lipid peroxidation is in basal conditions, the probe is excited at a wavelength of 581 nm and emits at a wavelength of 591 nm, observed with the PE filter. When marked lipid peroxidation occurs, there is a shift in the fluorescence of the fluorophore, which is excited at a wavelength of approximately 480 nm and emits at a wavelength of approximately 510 nm, observed with the FITC filter. This allows for a ratiometric analysis of lipid peroxidation.
7.- Conclusions: The conclusions describe the beneficial effects of OPE on cell life by reducing ROS generation and lipid peroxidation. Given that no specific compounds that are present in the extracts of OPE, I would recommend avoiding the inclusion of experiments that are not properly designed and do not contribute anything (HPLC). Thus, the discussion refers in general terms to the role of polyphenols and specifically to TYR, but without being able to state anything concrete, except to refer to its role as a scavenger of metals and oxidising species. Please avoid speculations.
Thank you for your observation. We have modified the Discussion (page 8, from line 9 to line 12).
Reviewer 2 Report
Comments and Suggestions for Authors
Edoardo Giuseppe Di Leo et al. reported here that the olive pomace extract acted as a potent antioxidant and prevented cells from ferroptosis. The topic is interesting and the result is promising, but a few major concerns must be addressed prior to the manuscript is acceptable for publication.
1. Since hydroxytyrosol (OH-Tyr) and tyrosol (Tyr) are consider the major components, why not just used the defined compounds for the same purpose?
2. Similar the 1, a direct comparison between OP extract and Tyr or OH-Tyr would be very helpful.
3. How consistent is the OP extract components? The author compared the HPLC with their own previous report, but the source of OP seems to be the same (from Produttori Agricoli Gardesani olive mill(Caprino Veronese, Verona, Italy) in 2022). Does other OP have similar components?
Some minor points:
1. some fonts in the figures are too small, for example in figure 3.
2. In figure 5A, OPE 50 seems to have no effect on GSH level of cells, but in 5B OPE 50 still showed a protection effect against RSL3, what's the mechanism behind this?
Author Response
Edoardo Giuseppe Di Leo et al. reported here that the olive pomace extract acted as a potent antioxidant and prevented cells from ferroptosis. The topic is interesting and the result is promising, but a few significant concerns must be addressed prior to the manuscript is acceptable for publication.
- Since hydroxytyrosol (OH-Tyr) and tyrosol (Tyr) are consider the major components, why not just used the defined compounds for the same purpose?
We thank the reviewer for this interesting observation. Actually, in the present study our aim was to take into consideration only the antioxidant effect of OPE in its entirety given it is the whole material that must be reutilized for sustainability. Therefore, in the Introduction, we have removed the phrase “OH-Tyr is a potent antioxidant with many biological effects [6], and its potential has been recognized by the European Food Safety Authority [7]” and the corresponding references. Accordingly, in the Discussion, we have shaded the sentences attributing all the antioxidant properties of OPE to OH-tyr and Tyr (page 8, from line 8 to line 12). In particular, we highlight the fact that OH-tyr and Tyr may play a major role in the antioxidant activity of OPE based on their highest concentration in the extract.
- Similar the 1, a direct comparison between OP extract and Tyr or OH-Tyr would be very helpful.
3. How consistent is the OP extract components? The author compared the HPLC with their own previous report, but the source of OP seems to be the same (from Produttori Agricoli Gardesani olive mill (Caprino Veronese, Verona, Italy) in 2022). Does other OP have similar components?
From our experience, olive pomace from different batches generally presents a similar qualitative profile of phenolic compounds. However, the relative abundance of individual components may vary depending on several factors, including cultivar (usually a blend is milled), fruit ripeness, and climatic conditions during the growing season. Given this inherent variability, and in order to reliably associate the observed biological activity with the composition of the extract, we chose to provide a detailed characterization of the phenolic compounds present in the sample used in this study (Under Materials and Methods, page 10, from line 5 to line 26).
Some minor points:
1. Some fonts in the figures are too small, for example, in Figure 3.
We correct font sizes in the Figures.
2. In figure 5A, OPE 50 seems to have no effect on GSH level of cells, but in 5B OPE 50 still showed a protection effect against RSL3, what's the mechanism behind this?
We suppose that in the presence of a consistent increase of oxidative stress and a huge decrease of GSH induced by RLS3, contrary to basal conditions, the key enzymes in the synthesis of GSH may have been potentiated by OPE.
Reviewer 3 Report
Comments and Suggestions for Authors
Dear Authors,
Thank you for the opportunity to review your manuscript. I commend your team for presenting a well-structured and scientifically relevant study on a new potent human cells ferroptosis inhibitor from olive pomace extract (OPE). The work is of scientific interest, particularly in the context of valorizing agricultural waste for nutraceutical or therapeutic applications. The study appears to combine chemical characterization, in vitro antioxidant assessment, and cell-based assays to elucidate the bioactivity of OPE. However, several aspects of the abstract could be improved for clarity, coherence, and scientific rigor.
Below are my comments and suggestions intended to strengthen the clarity, reproducibility, and scientific impact of your work:
- Abstract: The stated aim (“to assess the effect of an OP extract on RSL3-induced ferroptosis”) is appropriate but could be sharpened by explicitly linking the phenolic composition to the hypothesized bioactivity.
- Abstract: The conclusion appropriately connects the results to the proposed role of OPE as a ferroptosis inhibitor. However, the statement “OPE functions as a potent ferroptosis inhibitor” may be overly strong given that only in vitro data are reported. Consider moderating the language (e.g., “suggests potential as a ferroptosis inhibitor”).
- Abstract: It would help to briefly define RSL3 as a GPX4 inhibitor to provide context for non-specialist readers.
- Cell line names contain a potential typographical error: “TPH-1” likely refers to THP-1, and “HEIC” may be intended as HIEC (Human Intestinal Epithelial Cells?). Correcting these names is essential. Kindly ensure thoroughly in the entire manuscript.
- Please thoroughly check the entire manuscript for consistency in scientific writing. For example, chemical formulas such as hydrogen peroxide should be correctly formatted as Hâ‚‚Oâ‚‚, with the “2” written as a subscript.
Overall, this manuscript makes a valuable contribution to the field of valorizing agricultural waste for nutraceutical development. After the authors appropriately address the comments and suggestions provided, I recommend accepting this manuscript.
Author Response
Dear Authors,
Thank you for the opportunity to review your manuscript. I commend your team for presenting a well-structured and scientifically relevant study on a new potent human cells ferroptosis inhibitor from olive pomace extract (OPE). The work is of scientific interest, particularly in the context of valorizing agricultural waste for nutraceutical or therapeutic applications. The study appears to combine chemical characterization, in vitro antioxidant assessment, and cell-based assays to elucidate the bioactivity of OPE. However, several aspects of the abstract could be improved for clarity, coherence, and scientific rigor.
Below are my comments and suggestions intended to strengthen the clarity, reproducibility, and scientific impact of your work:
- Abstract: The stated aim (“to assess the effect of an OP extract on RSL3-induced ferroptosis”) is appropriate but could be sharpened by explicitly linking the phenolic composition to the hypothesized bioactivity.
We thank the reviewer for this important observation and accordingly, we explicitly add the phenolic content as a promoter of OPE antioxidant activity.
- Abstract: The conclusion appropriately connects the results to the proposed role of OPE as a ferroptosis inhibitor. However, the statement “OPE functions as a potent ferroptosis inhibitor” may be overly strong given that only in vitro data are reported. Consider moderating the language (e.g., “suggests potential as a ferroptosis inhibitor”).
We have modified as suggested.
- Abstract: It would help to briefly define RSL3 as a GPX4 inhibitor to provide context for non-specialist readers.
We have added as suggested
- Cell line names contain a potential typographical error: “TPH-1” likely refers to THP-1, and “HEIC” may be intended as HIEC (Human Intestinal Epithelial Cells?). Correcting these names is essential. Kindly ensure thoroughly in the entire manuscript.
Thank you for your suggestion, we have modified in HIEC in the entire text.
- Please thoroughly check the entire manuscript for consistency in scientific writing. For example, chemical formulas such as hydrogen peroxide should be correctly formatted as Hâ‚‚Oâ‚‚, with the “2” written as a subscript.
We have checked the manuscript and modified, accordingly.
Overall, this manuscript makes a valuable contribution to the field of valorizing agricultural waste for nutraceutical development. After the authors appropriately address the comments and suggestions provided, I recommend accepting this manuscript.
Round 2
Reviewer 1 Report
Comments and Suggestions for Authors
The manuscript has been significantly enhanced.
Reviewer 2 Report
Comments and Suggestions for Authors
All previous concerns have been clarified, and the manuscript is ready for publication.